# Using a Sensitivity Analysis and Spatial Clustering to Determine Vulnerability to Potentially Toxic Elements in a Semiarid City in Northwest Mexico

**Efrain Vizuete-Jaramillo** [1] , **Diana Meza-Figueroa** [2] , **Pablo A. Reyes-Castro** [3] **and Agustin Robles-Morua** [1,4,*]

1   Departamento de Ciencias del Agua y del Medio Ambiente, Instituto Tecnológico de Sonora, Obregón 85000, Mexico
2   Departamento de Geología, Universidad de Sonora, Hermosillo 83000, Mexico
3   Centro de Estudios en Salud y Sociedad, El Colegio de Sonora, Hermosillo 83000, Mexico
4   Laboratorio Nacional de Geoquímica y Mineralogía (LANGEM), Instituto de Geología, Universidad Nacional Autónoma de México, Ciudad de México 04510, Mexico
*   Correspondence: agustin.robles114592@potros.itson.edu.mx

**Abstract:** The Getis-Ord $G_i$* statistic clustering technique was used to create a hot spot exposure map using 14 potentially toxic elements (PTEs) found in urban dust samples in a semiarid city in northwest Mexico. The dust distribution and deposition in this city are influenced by the seasonal wind and rain from the North American Monsoon. The spatial clustering patterns of hot spots were used in combination with a sensitivity analysis to determine which variables most influenced the PTE hot spot exposure base map. The hot spots areas (%) were used as indicators of environmental vulnerability, and a final integrated map was selected to represent the highest vulnerability of PTEs with a 99% level of confidence. The results of the sensitivity analysis indicated that the flood zones and pervious and impervious zones were the most sensitive variables due to their weight in the spatial distribution. The hot spot areas were reduced by 60.4% by not considering these variables. The hot spot analysis resulted in an effective tool that allowed the combination of different spatial layers with specific characteristics to determine areas that present greater vulnerability to the distribution of PTEs, with impacts on public and environmental health.

**Keywords:** hot spot analysis; Getis-Ord $G_i$*; potentially toxic elements; kriging; GIS; urban dust

## 1. Introduction

Environmental pollution arising from atmospheric particles has become a prominent and serious public health problem in almost every city in the world. Rapid industrialization, agricultural development, and urban expansion have resulted in the increase in potentially toxic elements (PTEs) found in urban dust and soil samples [1–3]. The increased concentrations of PTEs found in dust particles and soil samples can be a source of chronic exposure to humans and may result in serious health risks [4–6]. There is currently evidence of a relationship between PTE exposure and chronic health effects, and it has been estimated that more than 70% of diseases caused by air pollution are associated with non-communicable diseases (NCDs) [4], especially in arid zones due to the large amount of dust found in the environment.

Arid conditions in urban areas are of special interest in air quality environmental studies [6] as a result of the extreme environmental conditions, such as temperatures exceeding 40 °C during summer [7,8], causing dry soil conditions. The dry climate and rapid urbanization in semiarid cities have led in some cases to reductions in the vegetation cover [8] and decreased natural barriers to dust deposition [9]. On the other hand, seasonal rain systems in semiarid cities are accompanied by periods of increasing winds and rain events that are of interest due to the sediment transport and resuspension processes that

occur on urban road surfaces [7,10]. The sediment that is collected by urban runoff accumulates dust that contains large amounts of pollutants, including PTEs [11]. These factors may enhance the resuspension, transportation, and redeposition of dust in different areas within cities [6,8]. As a result of the complex temporal dynamics of the weather (wind speed, relative humidity, air temperatures, rainfall), the movement and resuspension of dust can vary drastically within a season or from one to another. However, obtaining PTE estimates from field samples across large areas and multiple times a year is expensive and complicated. In recent years, clustering techniques have been proven to have the potential to reveal spatial patterns to identify environmental risks, particularly when combining multiple layers in the absence of extensive field data.

Clustering is a common method for statistical data analyses, and is used in many fields, including environmental sciences, computer science, and geographic information systems (GIS) [12,13]. The goal of clustering techniques is to partition a large amount of data into different subsets or groups [13–15]. However, grouping several variables can become a challenge for researchers due to the characteristics of spatial data, such as the scale, physical distance, and lack of homogeneity in the data [12,16]. Despite this, clusters can be useful to identify trends related to geographical phenomena if the data are converted and normalized in a heterogeneous format to simplify the analysis. Each clustering method may have advantages and disadvantages according to the study objective, size, type of data, number of clusters, and type of software used [13]. In most environmental studies, density-based clustering is the most common approach used because the data can be spatially represented on a physical level in several forms, using a raster format (grids cells or pixels) or vector format (point, lines, or polygons) [17]. The spatial data can then be compared and analyzed using various interpolation methods, such as kriging [18,19], in order to find where the data are closer together. The density of the data can be related to other explanatory variables. This is one of the main reasons why density-based clustering methods have been employed to better understand the relationships between multiple variables (layers) in many fields, such as global climate change, epidemic analyses, disease surveillance, population genetics, landscape ecology, earthquake analyses, and crime hot spot analyses [20–22].

The hot spot clustering analysis approach is a density-based method used to identify statistically significant areas by calculating $z$-scores based on the Getis-Ord $G_i$* statistic. The Getis-Ord $G_i$* statistic is calculated by comparing the sum of a point and its nearest neighbors to the sum of all points in a given study area. The hot spot analysis method has been widely used to identify environmental and public health risks and social problems, and even to identify traffic jams. Some notable studies involving PTEs include the study by Kim and Choi (2017,2019), who carried out a hot spot analysis to identify areas of vulnerability to PTE exposure using data obtained from portable X-ray fluorescence images in South Korea [23,24]. Similarly, Xu et al. (2021) used a hot spot analysis to determine the spatial patterns of PTEs using 6862 topsoil samples in Northern Ireland [25].

There are many other applications where the hot spot technique has been used. For example, Kumari and Pandey (2020), Rossi and Becker (2019), Said et al., (2017), and Zahran et al. (2020) used a hot spot analysis to identify fire risks using point data from fire reports and satellite data [26–29]. Zahran et al. (2019) carried out a spatial analysis using the hot spot tool to identify areas of greater vulnerability to traffic accidents using data from accident reports provided by the police department [30]. Lu et al. (2019) used a hot spot analysis to evaluate potential landslide risks in Italy [31]. During the global pandemic due to the coronavirus, the hot spot analysis approach has been of great importance in determining the areas of greatest risk from the virus. Kuznetsov and Sadovskaya (2021) and Shariati et al. (2020) used a hot spot analysis to understand the spatial distribution of the coronavirus disease at regional and global scales [32,33]. All of these studies demonstrate the wide range of applications where clustering techniques can be used, particularly when using the Getis-Ord $G_i$* tool. However, despite developing important information about vulnerable locations to different risks, most of these studies have focused on evaluating

independent variables individually, without considering the relationships, combinations, or influence of other variables that could increase or decrease the vulnerability to a particular problem (percent influence or confidence intervals).

To our knowledge, the studies where the combination of two or more variables are considered are still limited and few studies have quantified the influence of each variable on the final vulnerability or risk map. Most studies generated statistical difference metrics at the urban scale, but did not consider metrics within the urban areas or using the spatial resolution of the variables used in the analysis. For example, Lee and Khattak (2019) and McClintock (2012) used *z*-scores of hot spot analyses to identify the differences between different variables at the urban scale [34,35]. McEntee and Ogneva-Himmelberger (2008) carried out a hot spot analysis for the exposure of diesel particulates using a *t*-test to determine the significant differences between the influence of lung cancer and asthma cases [36]. Several other studies have determined significant differences (*p*-values) between variables used in hot spot analyses aggregated at the urban scale [37,38]. Navarro-Estupiñan et al. (2020) created a heat risk map and used the percentages of areas being affected to demonstrate the areas that were more vulnerable [39]. However, these authors did not evaluate the impacts of each layer separately.

Despite the relevance of these studies, few studies have quantitatively demonstrated through sensitivity analyses how each of the variables can influence the final outcome of the spatial clustering of hot or cold spots in the study areas. On the other hand, studies of the vulnerability to PTEs in semi-arid areas are even less known. This research has two main objectives: (i) to integrate multiple variables related to physical and public health characteristics to identify areas that are vulnerable to PTEs; (ii) to determine the influence that each variable has on the spatial distribution of vulnerable areas (hot spots) through a sensitivity metric analysis. The hypothesis being tested here is that the incorporation of multiple variables or spatial layers can provide a more robust measure of vulnerability rather than simply relying on the distribution of the highly variable areas of urban dust deposition that are dependent on expensive field studies. The results of this work will allow government planning and public health agencies and developers to identify variables that require greater attention in urban planning to reduce vulnerability in cities through sensitivity analyses.

## 2. Materials and Methods

### 2.1. Dust in the Study Area and Its Relationship with the Monsoon

This study was conducted in the city of Hermosillo, which is located in northwestern Mexico within the Sonoran Desert (Figure 1). It is a semiarid urban area characterized primarily by commercial, agricultural, and industrial activities [40]. Over the last two decades, the city has experienced a very rapid population growth (4.6%) from 406,417 inhabitants in 1990 to 855,563 in 2020 [41], caused primarily to migration due to the city's economic opportunities. The rapid growth of the city, coupled with a lack of long-term planning for urban infrastructure, has resulted in serious consequences for air quality [42].

Dust particles are produced naturally by wind erosion in arid and semiarid regions around the world [43]. However, when combined with heavy metals emanating from anthropogenic sources, the two sources pose a threat to public health due to the high concentration of PTEs [6,44]. In the city of Hermosillo, the movement of local air masses linked to the local topography and building distribution, in addition to the seasonal evolution of the North American Monsoon (NAM), has resulted in a unique movement of urban dust. The eastern side of the city is surrounded by hills, with a north-north-west (NNW) to south-south-west (SSE) outlook and prevailing wind directions that are from west to east. Typically, before the monsoon arrives, strong wind gusts and higher air temperatures occur. Then, heavy rains begin during the months of June and July. This pattern of the NAM occurs over southern Mexico and rapidly spreads northward along the western slopes of the Sierra Madre Occidental [45,46]. The typical monsoon onset dates

range from early June in southwestern Mexico to early to mid-July in Sonora, Arizona, and New Mexico.

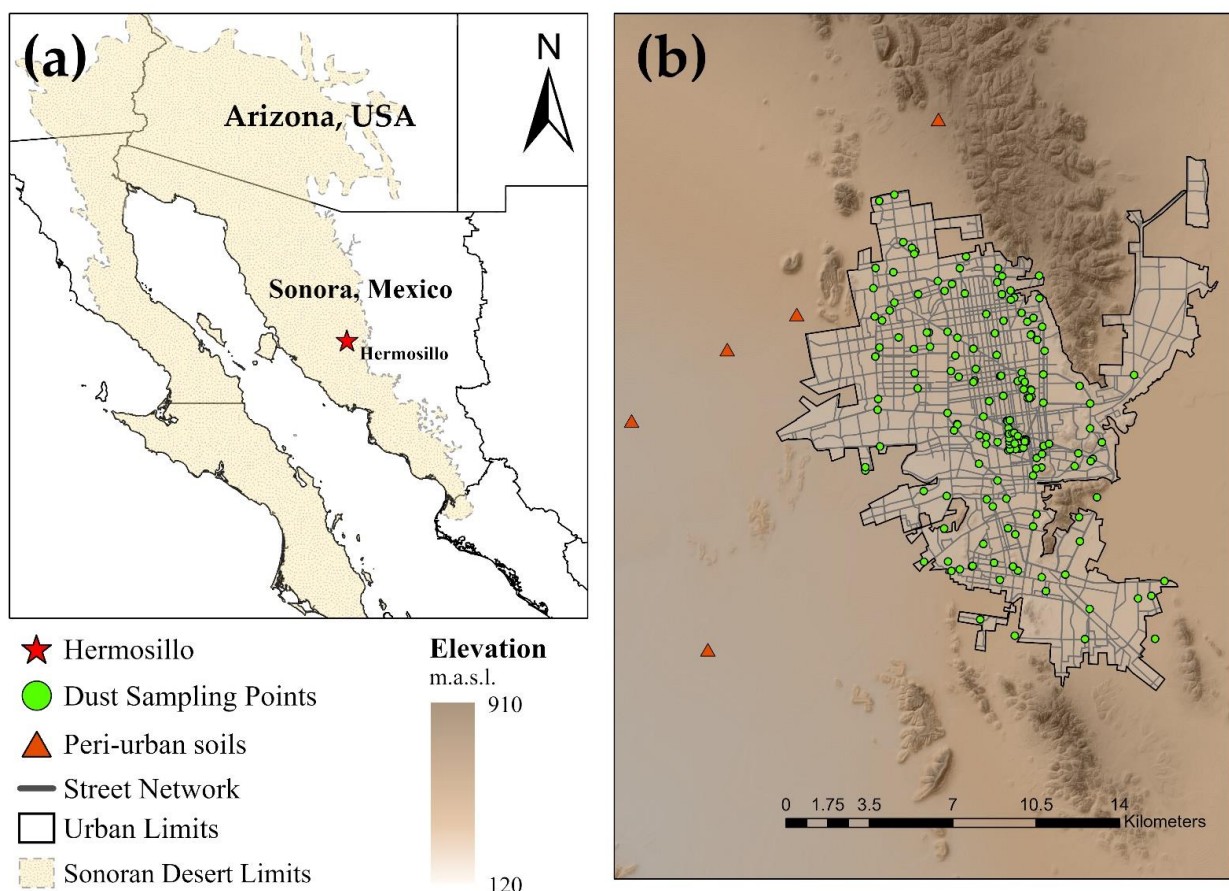

**Figure 1.** (**a**) Location of Hermosillo within the state of Sonora. (**b**) Location map of the sampling points. The gray lines are high-traffic roads. Dust samples were collected within the urban limits and soil samples were collected outside the urbanized area (peri-urban soil samples) as a control site.

In any location, the onset of the monsoon is usually sudden, and the climate changes abruptly as a result of the extreme windy conditions. The temperature also changes from relatively hot and dry conditions to relatively cool and rainy conditions [47]. In the city of Hermosillo, the monsoon exhibits a similar behavior to that of the larger scale of the NAM. The weather conditions registered in the city range from 35 to 49 °C during summer and from 8 to 5 °C during winter, while the precipitation ranges from 75 to 300 mm yr$^{-1}$. During the monsoon season, intense rainfall over a short duration occurs in the city and surrounding areas, causing floods with high-velocity surface runoff that flows into the city via the street network and following the directions of north to south-west and north to west due to the lack of drainage systems. The high-velocity surface runoff carries sediment from the hills located in the northern part of the city, which are later resuspended by traffic and the windy conditions prior to the start of the following rainy season [6]. The intense rainfall also has the potential to break the soil crust, causing soil erosion, which in turn releases dust [48]. In recent years, the enhancement of dust particles associated with the monsoon season has been reported by different researchers [2,6,49].

### 2.2. Dust Sampling and Analysis

The sampling sites were selected to cover various areas in the city of Hermosillo, including residential areas with low, medium, and high population density levels; industrial areas; public parks; heavy and low traffic areas; and commercial areas. The

settled dust samples (*n* = 170) were collected from the streets by sweeping an area measuring two square meters using a polyethylene brush, tray, and containers as suggested by Meza-Figueroa et al. [6,40]. The settled dust samples were collected throughout 2014. The geochemical background values were obtained from four peri-urban soil samples that were collected outside of the urbanized area (Figure 1).

The dust and soil samples were weighed and oven-dried at 40 °C for 24 h. Afterward, the samples were mechanically sieved with mesh < 325, corresponding to a grain size of <45 μm; this fraction represents a particle size that can be easily resuspended by the wind [2,6]. This size was chosen because the finer dust particles generally contain higher concentrations of PTEs [40]. Thirty-six PTEs were analyzed from the dust samples using a portable X-ray fluorescence analyzer (Niton XL3t Analyzer, Thermo Scientific, Inc., Waltham, MA, USA) according to method 6200 [50] from the United States Environmental Protection Agency (USEPA). An exhaustive data analysis was performed using descriptive statistics to eliminate elements with a small amount of data. Depending on the interpolation methods, the number and distribution of points (data) play an important role in increasing or decreasing the interpolation error. A larger amount of data will considerably decrease the error during the interpolation [51]. For the same reason, fourteen PTEs (Ba, Ca, Cr, Cu, Fe, K, Nb, Pb, Sr, Ti, Th, V, Y, and Zn) were selected because they met the necessary requirements (greater amount of data) to integrate them into the final analysis. The detection limits were 28, 20, 7, 6, 20, 30, 2, 2, 2, 10, 2, 8, 2, and 4 mg·kg$^{-1}$, respectively, for each of the PTEs.

### 2.3. Spatial Variability of PTEs, Physical and Public Health Variables Associated with Dust Distributions

Multiple variables were selected due to their relationships with the movement of and exposure to urban dust. The physical and public health variables associated with urban dust were prepared and normalized to be integrated with each other and to evaluate their influence on the vulnerability associated with PTEs (vulnerability map). Spatial distribution maps of the PTEs, flood zones, pervious and impervious zones, and industrial zones, as well as an age-adjusted NCD mortality rate map for the city of Hermosillo, were used to elaborate a single map that represented the vulnerability to the PTEs contained in urban dust. These variables were also used to evaluate the influence on the changes in vulnerable areas related to PTEs through a sensitivity analysis. The sensitivity analysis was designed to quantify the impacts of each layer independently and in combination with others on the vulnerable areas (hot spots) in terms of the percentages for each variable. All possible combinations between the variables were assessed to evaluate changes in vulnerable areas in the study area. Figure 2 shows the flowchart that describes the conceptual model used to create every layer and the experimental design of the sensitivity analysis, which resulted in a single map of the combined vulnerability.

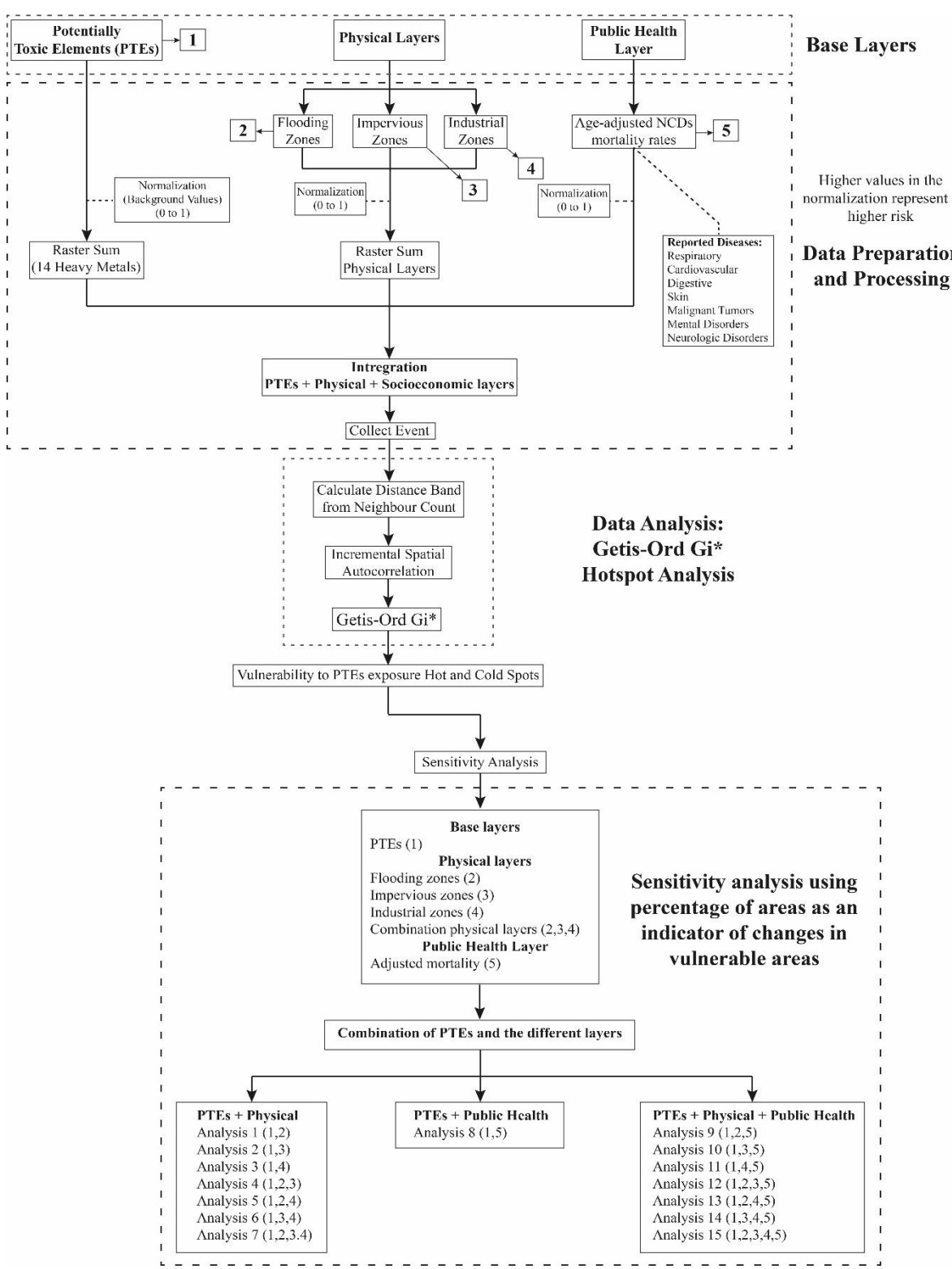

**Figure 2.** Conceptual model of the work conducted in our study, which included the preparation and processing of data, the creation of hot spot maps (Getis-Ord $G_i^*$), and a subsequent sensitivity analysis of the vulnerability to PTE exposure.

### 2.4. PTE Exposure Map

In order to obtain spatial distribution maps of Ba, Ca, Cr, Cu, Fe, K, Nb, Pb, Sr, Ti, Th, V, Y, and Zn, a kriging interpolation process was conducted using the analysis of each element

for each sample of urban dust. The spatial maps were created using ArcGIS Pro 2.8. Kriging is a spatial interpolation and estimation method that is widely used in meteorology, mining, geology, environmental studies, and agriculture applications [19,52]. The advantage of using interpolation methods such as kriging is being able to estimate the value of a variable in one or several unknown sites, considering the known values close to the sites of interest, which is called interpolation [19,53].

A descriptive statistics assessment and analysis of variance were performed prior to conducting the kriging interpolations to verify whether the data presented a normal or non-normal distribution according to the Shapiro–Wilk test (Table 1).

**Table 1.** The descriptive statistical analysis.

| Element | Symbol | N | Min | Max | Average | Std. Dev | Normal Distribution Test | |
|---|---|---|---|---|---|---|---|---|
| | | | | **mg·kg$^{-1}$** | | | **p \*** | |
| Barium | Ba | 170 | 148.7 | 979.2 | 508.6 | 128.9 | 0.076 | normal distribution |
| Calcium | Ca | 169 | 9861.0 | 124,682.0 | 42,718.0 | 19,332.0 | <0.010 | non-normal distribution |
| Chromium | Cr | 139 | 112.6 | 273.5 | 165.3 | 26.7 | <0.010 | non-normal distribution |
| Copper | Cu | 165 | 29.0 | 13,264.0 | 363.0 | 1315.0 | <0.010 | non-normal distribution |
| Iron | Fe | 170 | 17,416.0 | 82,564.0 | 36,350.0 | 8787.0 | <0.010 | non-normal distribution |
| Potassium | K | 170 | 15,707.0 | 26,913.0 | 20,427.0 | 1917.0 | <0.010 | non-normal distribution |
| Niobium | Nb | 170 | 9.3 | 34.1 | 19.3 | 3.7 | <0.010 | non-normal distribution |
| Lead | Pb | 170 | 15.2 | 979.8 | 100.9 | 95.5 | <0.010 | non-normal distribution |
| Strontium | Sr | 169 | 258.0 | 659.6 | 402.7 | 64.7 | <0.010 | non-normal distribution |
| Titanium | Ti | 170 | 2210.6 | 8408.9 | 4820.4 | 930.1 | <0.010 | non-normal distribution |
| Thorium | Th | 167 | 11.2 | 76.7 | 29.4 | 11.3 | <0.010 | non-normal distribution |
| Vanadium | V | 109 | 153.8 | 392.9 | 242.1 | 38.6 | <0.010 | non-normal distribution |
| Yttrium | Y | 170 | 13.8 | 55.8 | 30.4 | 6.4 | <0.010 | non-normal distribution |
| Zinc | Zn | 169 | 63.5 | 1283.7 | 243.6 | 181.4 | <0.010 | non-normal distribution |

\* Significance level 0.05.

For each element, the variograms were calculated as an input parameter based on the kriging interpolation. A variogram is a tool that allows one to analyze the spatial behavior of a variable over a defined area, obtaining, as a result, an experimental variogram that reflects the maximum distance and the way in which one point influences another point at different distances [53]. The geostatistical data were analyzed using the program GS+ (Gamma Design Software, 2006), followed by an exploratory data analysis and a spatial autocorrelation process as suggested by Cortés et al. (2017) and Delgado et al. (2018) [54,55]. Finally, the variogram calculations were carried out using Equation (1):

$$\gamma(h) = \frac{1}{2n(h)} \sum_{i=1}^{n(h)} [z(x_i) - z(x_i + h)]^2 \tag{1}$$

where $h$ is a vector of two separate sampling sites (also called the step length), $z(x_i)$ is equal to the value of the PTEs at location $x_i$, $z(x_i + h)$ is equal to the value at location $x_i + h$, $\gamma(h)$ is equal to the variogram for the distance h between values $z(x_i)$ and $z(x_i + h)$, and $n(h)$ is equal to the number of pairs of values separated by $h$ [53,56].

The variograms were fitted to the linear, exponential, gaussian, or spherical models depending on the characteristics of each dataset. Each model was applied to an interpolation method (Ordinary Kriging) to obtain maps of the distribution of each element (Ba, Ca, Cr, Cu, Fe, K, Nb, Pb, Sr, Ti, Th, V, Y, and Zn) using the dust data (point data) collected during 2014 in Hermosillo.

Covariates were added to the kriging interpolation model for a better representation of the spatial distribution. A 15-meter digital elevation model provided by the National Institute of Statistics and Geography (INEGI) was used to represent the roughness of the terrain of the city and its surroundings. In addition, a wind speed and direction raster from 2014 was obtained from the North American Land Data Assimilation System (NLDAS) to

represent the potential dispersion of the PTEs. A total of ten classes of concentrations were defined for each element. The maps were created with ArcGIS Pro 2.8 (ESRI Inc, Redlands, CA, USA).

The spatial distribution maps of the 14 elements were normalized from 0 to 1, where 0 represents the background value (Hermosillo soil concentration) (See Table 2) and 1 is the maximum value found for each element. Once each map was normalized, the 14 maps were summed and averaged to obtain an integrated exposure map of PTEs in Hermosillo. This integrated map was considered as the main input described in the flowchart (Figure 2).

**Table 2.** Elemental contents in soil samples from around the world and the soil samples from Hermosillo.

| Element | Symbol | Global Concentration (Soil) | | | Hermosillo Soil Concentration |
|---|---|---|---|---|---|
| | | Median | Min | Max | Average |
| | | mg·kg$^{-1}$ | | | mg·kg$^{-1}$ |
| Barium | Ba | 500 | 100 | 3000 | No data |
| Calcium | Ca | 15,000 | 700 | 500,000 | 20,715.75 |
| Chromium | Cr | 70 | 5 | 1500 | 53.46 |
| Copper | Cu | 30 | 2 | 250 | No data |
| Iron | Fe | 40,000 | 2000 | 550,000 | 20,973.75 |
| Potassium | K | 14,000 | 80 | 37,000 | 21,095.8 |
| Niobium | Nb | 10 | 6 | 300 | 13.06 |
| Lead | Pb | 35 | 2 | 300 | 20.9 |
| Strontium | Sr | 250 | 4 | 2000 | 268.26 |
| Titanium | Ti | 5000 | 150 | 25,000 | 3265.1 |
| Thorium | Th | 9 | 1 | 35 | 19.88 |
| Vanadium | V | 90 | 3 | 500 | 75.84 |
| Yttrium | Y | 40 | 10 | 250 | 20.68 |
| Zinc | Zn | 90 | 1 | 900 | 52.84 |

Global concentration data were obtained from Sparks [57] and the Hermosillo soil concentrations were obtained from peri-urban soil data.

### 2.5. Environmental Variables

Other relevant physical layers were considered as an input to the model due to the role that they play in the transportation and distribution of urban dust containing PTEs in Hermosillo. A total of three maps were used as some of the environmental variables considered in our analysis. The official map of flooding in the city was obtained from the municipal institute of urban planning (IMPLAN). Flood zones are important due to the transport of sediment in the streets that accumulate there because of the city's lack of a storm drainage system. During the flood events, the accumulation of sediment occurs in different areas of the city. The flood zones map was classified from 0 to 1, where 1 represents flooded areas and 0 non-flooded areas (Figure 3a).

A map of pervious and impervious zones was created from the official land use map of Hermosillo [58]. The land uses were classified according to their permeability, ranging from 0 to 1, with increments of 0.1 (Figure 3b), where green areas and natural soil areas are represented by values of 0 due to the easy permeability of water in the soil. The values of 1 represent concrete, paved streets and sidewalks, high-risk industries, and urban infrastructure in general, representing impermeable areas. High-impermeability conditions limit the infiltration of water, meaning the PTEs are available for resuspension from the surfaces of these impermeable areas and can also move superficially via advection with pulses of urban runoff. Similarly, the industrial zones map was created from the official land use map of Hermosillo [58], and the urban zones were classified as 0, low-risk industries as 0.25, commercial industry risk as 0.5, medium-risk industries as 0.75, and high-risk industries as 1 (Figure 3c).

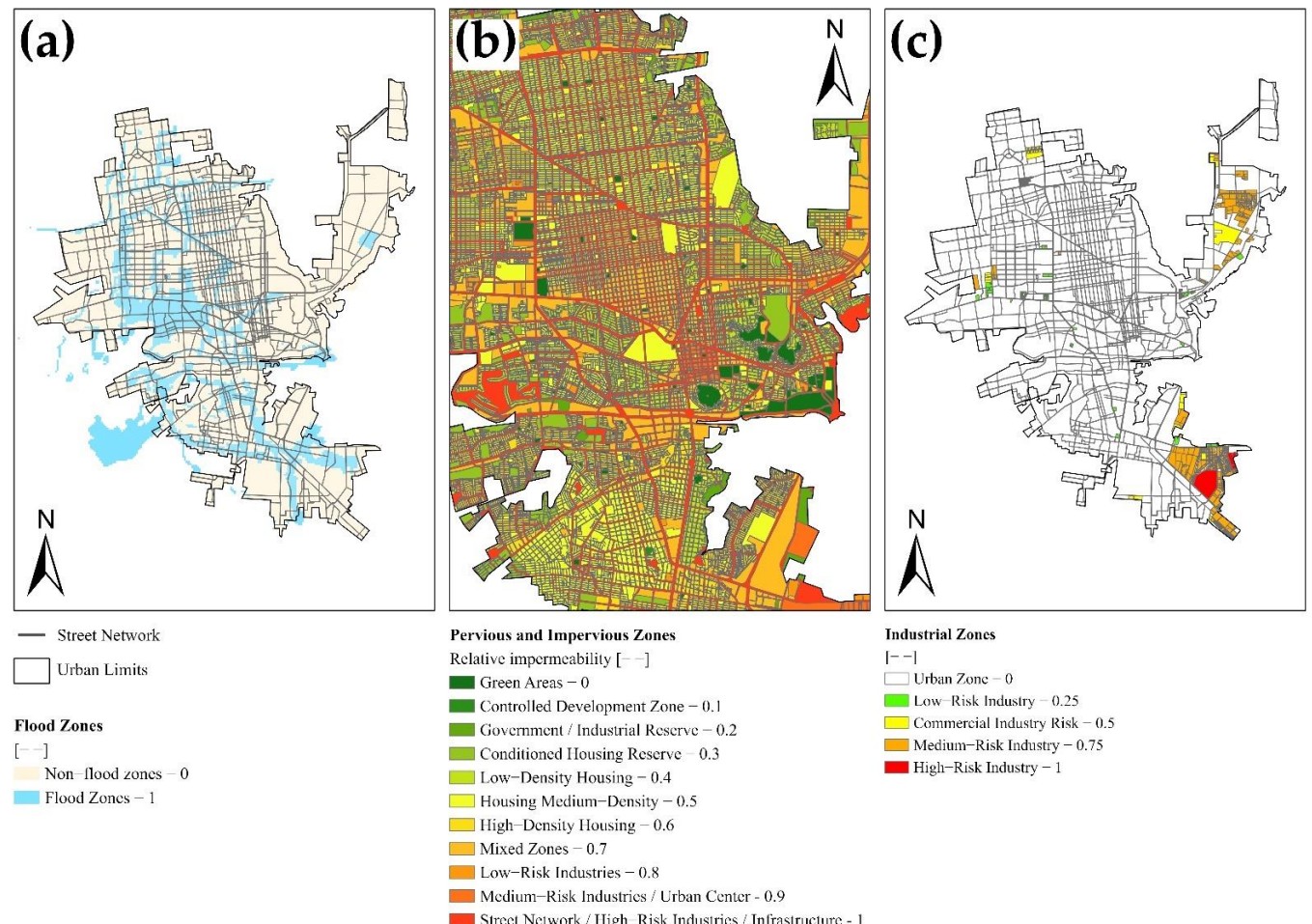

**Figure 3.** (**a**) Official flood zones map in Hermosillo. (**b**) Pervious and impervious zones in Hermosillo. The image was magnified to appreciate the details of the pervious and impervious zones. (**c**) The industrial zone map represents the different industries and their contributions to air pollution.

*2.6. Public Health Layer Data*

The public health map was obtained from the work published by Reyes-Castro (2019) [59]. In this work, the spatial distribution of the intra-urban mortality by areas of residence and the association with marginalization and population ageing in Hermosillo during 2013–2016 were mapped. Three groups of causes of death were determined, namely communicable diseases, non-communicable diseases, and external causes of morbidity and mortality. In this work, non-communicable disease data were used (Table 3) and mapped using an age-adjusted NCD mortality rate, which may be related to exposure to PTEs. Reyes-Castro (2019), at the level of the basic geostatistical areas (BGAs), georeferenced 97.6% of the total deaths that occurred in the study period and estimated the age-adjusted mortality rate at the BGA level for each cause of death [59]. The rates were presented as deaths per 10,000 inhabitants. Finally, the rates were smoothed using the empirical Bayesian method to correct the variance instability of the small BGAs with low populations (Figure 4a).

**Table 3.** Causes of death (Group II. non-communicable diseases) reported from 2013 to 2016 from non-communicable diseases in Hermosillo.

| Causes of death (Group II. non-communicable diseases) |
|---|
| Malignant tumors. |
| Other tumors. |
| Mellitus diabetes. |
| Endocrine, metabolic, hematological, and immunological diseases. |
| Mental disorders and diseases of the nervous system. |
| Diseases of the sense organs. |
| Cardiovascular diseases. |
| Respiratory diseases. |
| Digestive diseases. |
| Diseases of the genitourinary system. |
| Skin diseases. |
| Diseases of the musculoskeletal system. |
| Congenital anomalies. |
| Diseases of the mouth. |

Data for non-communicable diseases in Hermosillo were obtained from Reyes-Castro [59].

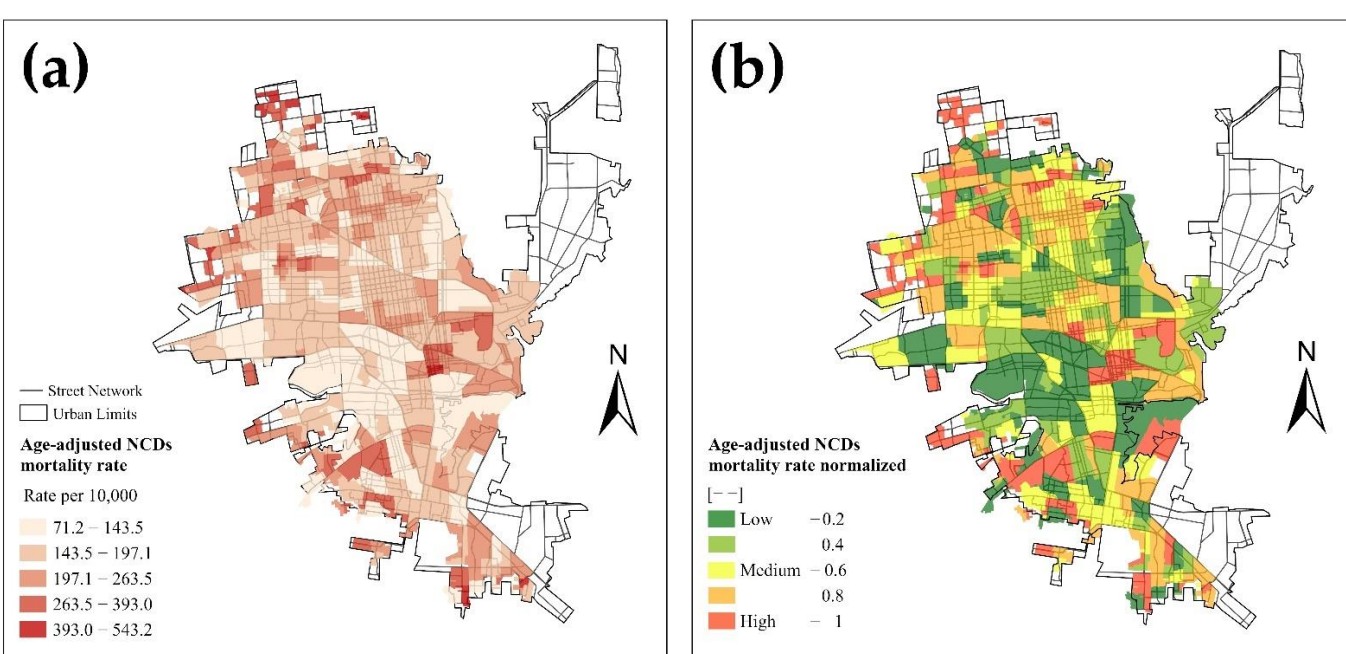

**Figure 4.** (**a**) Age-adjusted mortality rates using data for non-communicable diseases (NCD) in Hermosillo mapped by Reyes-Castro. (**b**) Age-adjusted NCD mortality rates normalized for the hot spot analysis [59].

The age-adjusted NCD mortality rates published by Reyes-Castro (2019) show five levels—71.2 to 134.6, 134.7 to 164.2, 164.3 to 189.8, 189.9 to 242.0, and 242.1 to 543.2 [59]. The range was converted and normalized from 0.2 to 1, starting with the first level, with an increase of 0.2, until reaching the last level, where the lowest values represent low mortality rates and the highest values of the index represent high mortality rates (Figure 4b).

### 2.7. Critical Parameters for Hotspots

After all spatial layers were normalized, they were used to conduct a hot spot analysis (the first hotspot map was the PTE exposure map that integrated the 14 elements). However, the critical parameters must be calculated as input sources for the different analyses. The beginning distance, distance increment, and distance band were required to be set before the hot spot analysis could be run (Table 4). These parameters were calculated following

the steps proposed by Said et al. [28]. The calculated distance band from the neighbor count tool was used to determine the minimum, average, and maximum distances that each layer has to at least one neighbor. The resulting maximum distance of each analysis was used as the beginning distance. The average distance obtained from this tool was applied as a distance increment in each analysis. In this case, the increment distance was made smaller in an attempt to capture a more noticeable peak. Thus, the distance at which the points were most clustered was determined.

**Table 4.** Critical parameters for the Getis-Ord $G_i^*$ statistics calculated for each analysis.

| Analysis | Average Distance | Maximum Distance | Distance Band |
|---|---|---|---|
| **PTEs layer** | | | |
| Potentially Toxic Elements (PTEs) | 149 m | 686 m | 2892 m |
| **Physical layers** | | | |
| Flood zones | 518 m | 3287 m | 7946 m |
| Pervious and Impervious zones | 43 m | 634 m | 1366 m |
| Industrial zones | 61 m | 1565 m | 1565 m |
| Flood zones + Pervious and Impervious Zones + Industrial zones | 99 m | 784 m | 1874 m |
| **Public health layer** | | | |
| Age-adjusted NCD mortality rate | 422 m | 1508 m | 2351 m |
| **Combination of PTEs and the different layers** | | | |
| **PTEs + Physical** | | | |
| Analysis 1 | 159 m | 630 m | 2048 m |
| Analysis 2 | 105 m | 404 m | 2527 m |
| Analysis 3 | 145 m | 686 m | 3574 m |
| Analysis 4 | 102 m | 457 m | 1929 m |
| Analysis 5 | 137 m | 615 m | 1982 m |
| Analysis 6 | 104 m | 388 m | 3209 m |
| Analysis 7 | 101 m | 457 m | 1928 m |
| **PTEs + Public Health** | | | |
| Analysis 8 | 132 m | 500 m | 2632 m |
| **PTEs + Physical + Public Health** | | | |
| Analysis 9 | 124 m | 572 m | 696 m |
| Analysis 10 | 98 m | 367 m | 2302 m |
| Analysis 11 | 131 m | 500 m | 2632 m |
| Analysis 12 | 96 m | 379 m | 854 m |
| Analysis 13 | 124 m | 572 m | 695 m |
| Analysis 14 | 98 m | 367 m | 2302 m |
| Analysis 15 | 95 m | 379 m | 854 m |

To determine the distance band, an incremental spatial autocorrelation calculation was performed. This tool measures the degree of clustering of data in space at an increasing distance. On the other hand, this tool creates a graph that can be used as an appropriate scale of analysis or as a distance band. The peak in the graph indicates the distance at which the clustering is more pronounced. The clustering distance was used in the Getis-Ord $G_i^*$ analysis as a distance band. These parameters were calculated based on a sensitivity analysis carried out among all possible combinations between the PTE map, physical maps, and public health map (See Figure 2).

### 2.8. Hotspot Analysis: Getis-Ord $G_i^*$

The Getis-Ord $G_i^*$ hotspot analysis method was used for the identification of PTE vulnerability hotspots in the study area. This hotspot analysis utilizes $G_i^*$ statistics that can be calculated as follows:

$$G_i^* = \frac{\sum_{j=1}^n w_{ij} x_j}{\sum_{j=1}^n x_j} \qquad (2)$$

where $G_i*$ is the spatial autocorrelation (spatial dependency) statistics of an event $i$ over $n$ events, the term $x_j$ defines the magnitude of variable $x$ at events $j$ over all $n$, and the term $w_{ij}$ defines the weight value between the events $i$ and $j$ that represents their spatial interrelationship. The $G_i*$ statistics consider the magnitude of each feature in the dataset in the context of its neighbors' values. The local sum of a feature and its neighbors is compared accordingly to the sum of all features. If there is a significant difference between the local sum and the expected local sum, where the difference is too large due to randomness, a statistically significant $z$-score is the result [26,28].

To further elaborate, the Getis-Ord $G_i*$ hotspot analysis determines where the features with high and low $z$-scores and $p$-values tend to form a cluster in the study area. The analysis tool calculates the $z$-score and $p$-value for each feature, which can help to indicate the cold and hot spots of events. The $z$-score output represents the statistical significance of the clustering for a specified distance, whereas the $p$-value indicates the probability that the observed spatial pattern was created by a random process.

### 2.9. Sensitivity Analysis Calculations

Once all of the hotspot maps were obtained, the sensitivity analysis to determine the vulnerability maps to PTE exposure was carried out. The areas of each cold and hot spots (90, 95, 99%) on the maps were calculated in units of km$^2$. To calculate the area percentage for the different analyses, the following equation was used:

$$\%_{hs} = \frac{\sum Area \ G_i\_Bin_i}{TA} * 100 \tag{3}$$

where $\%_{hs}$ is the percentage of each confidence level (cold spots at 90, 95, and 99% confidence and hot spots at 90, 95, 99% confidence), Area $G_i\_Bin_i$ represents the sum of areas at each confidence level, and TA is the total area of the city. The calculation of the areas of the cold and hot spots was considered to determine the influence of each layer in the global analysis.

## 3. Results

### 3.1. Geostatistical Analyses of PTEs

For the spatial distribution maps of PTEs to be created, variograms (models) and nuggets must be obtained. Models were obtained for each element, such as Gaussian, spherical, exponential, and linear models. The nugget effect was also calculated, with the geostatistical analysis results shown in Table 5. These parameters are critical for the kriging interpolation methodology used to create spatial distribution maps.

**Table 5.** Geospatial analysis.

| Element | Symbol | Model | $R^2$ | Structural Variance | Nugget |
|---------|--------|-------|-------|---------------------|--------|
| | | Variogram | | % | |
| Barium | Ba | Spherical | 0.95 | 93.4 | 11,342.8 |
| Calcium | Ca | Spherical | 1.10 | 68.7 | 124,792,759.6 |
| Chromium | Cr | Exponential | 1.00 | 88.5 | 647.6 |
| Copper | Cu | Gaussian | 1.47 | 89.3 | 706,485.8 |
| Iron | Fe | Gaussian | 1.21 | 63.5 | 48,187,731.4 |
| Potassium | K | Gaussian | 1.09 | 67.8 | 2,154,495.0 |
| Niobium | Nb | Exponential | 1.05 | 71.3 | 11.3 |
| Lead | Pb | Linear | 0.92 | 33.3 | 8570.2 |
| Strontium | Sr | Exponential | 1.11 | 71.3 | 1412.6 |
| Titanium | Ti | Spherical | 1.04 | 62.8 | 535,356.4 |
| Thorium | Th | Gaussian | 0.97 | 59.1 | 114.2 |
| Vanadium | V | Gaussian | 1.31 | 52.4 | 1019.6 |
| Yttrium | Y | Gaussian | 1.05 | 60.9 | 32.6 |
| Zinc | Zn | Gaussian | 1.33 | 87.5 | 12,187.5 |

The parameters were calculated using the program GS+. All elements correspond to an isotropic variogram.

### 3.2. Spatial Distribution of PTEs

Figure 5 represents an example of three individual elements used for the spatial distribution in the PTE analysis. In this case, Pb, Cr, and V were selected because we could compare these elements with previous studies already published in the same study area and because of their potential public health effects [6,8,49]. Figure 5a shows the distribution of Pb in Hermosillo, which we argue could be the result of vehicle emissions, since Pb is used as an additive in gasoline in Mexico [40]. High concentrations were found in the east of Hermosillo. In the case of Figure 5b, a significant concentration of Cr is shown in the center of the city, which could be associated with metal-based traffic paint on the streets and playground paint [6]. The last element shown in Figure 5c represents the spatial distribution of V, which is considered a marker of heavy fuel oil combustion. Vanadium is mainly used in the steel manufacturing, aircraft, and armament industries. Vanadium is also widely used in burning fossil fuels, smelting, and mining [40,60,61]. The Figure 5c shows clustering areas of V in the north and southwest of the city, which may be associated with commercial and industrial activities in the city, such as the use of fuels, the locations of cement plants, and the steel industry [40].

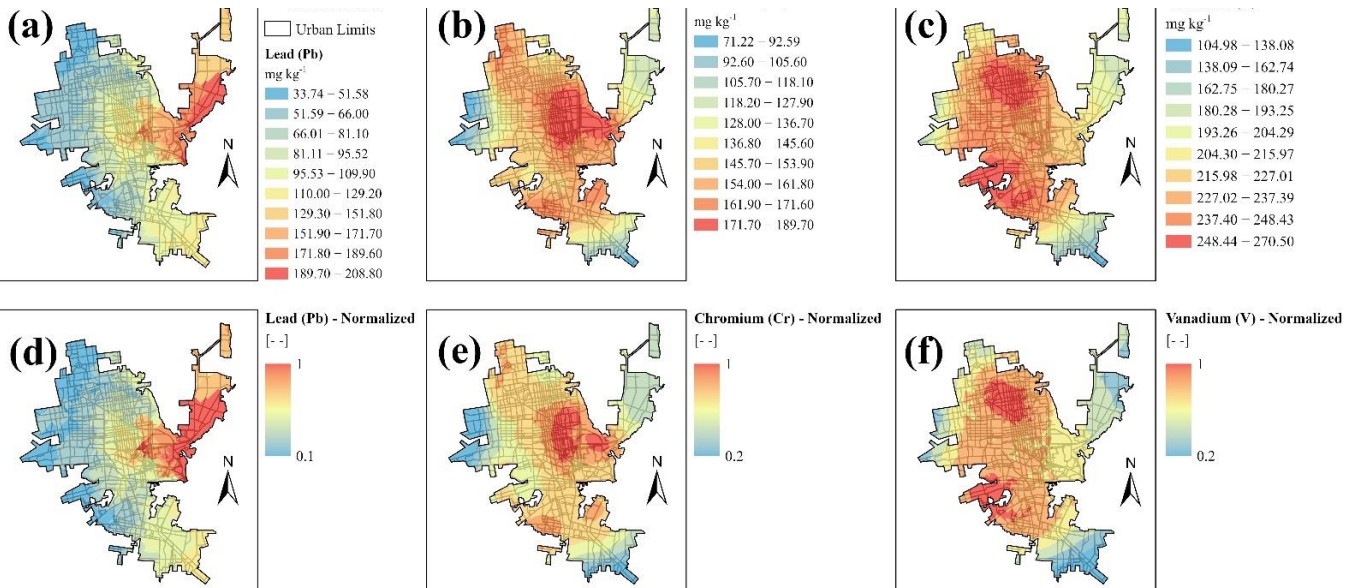

**Figure 5.** Spatial distribution map using kriging as an interpolator for (**a**) lead, (**b**) chromium, and (**c**) vanadium. Normalized map using the background value of 0 and the maximum values for (**d**) lead, (**e**) chromium, and (**f**) vanadium.

Figure 5d–f shows the normalization maps of Pb, Cr, and V obtained by eliminating the background values of each element. In other words, all concentrations shown mean that the origin is not natural and they did not come from rocks or soil. In the case of Pb, Cr, and V, the background values are 20.9, 53.4, and 75.8 mg·kg$^{-1}$, respectively, and the minimum values of each element are above the background values. The maps of all 14 elements used in this study are available in the Supplementary Materials section.

Once the interpolation maps of each of the 14 elements were generated, they were normalized and then summed to obtain an integrated exposure map of PTEs in Hermosillo (Figure 6). The high values in the range indicate the greater presence of all elements. Figure 6 represents the areas within the city limits with at least one of the 14 elements present above their background concentration value. This was considered as the base map of PTE exposure in Hermosillo (item # 1 in the conceptual diagram).

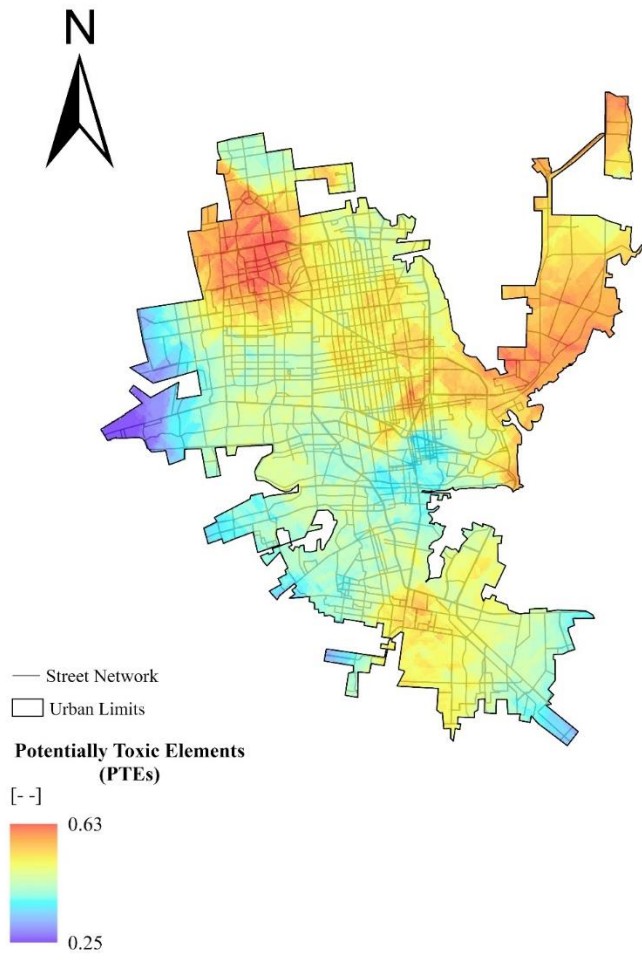

**Figure 6.** Spatial distribution of PTEs in Hermosillo. This figure represents the spatial distribution of 14 elements. The highest values of this normalized map represent the presence of most of the elements in a given site.

### 3.3. Hot Spot Vulnerability Maps

The Getis-Ord $G_i^*$ hot spot tool was used to obtain a total of 21 maps. Figure 7 shows the result of the hot spot analysis used for base maps of the PTEs, flood zones, pervious and impervious zones, industrial zones, and a combination of all of the physical layers. The areas of highest vulnerability are those represented by a 99% confidence value. Figure 7a shows the most vulnerable areas of exposure to the 14 PTEs in the city. Due to the great presence of these elements in the north of the city, an important cluster can be observed in that area. In the case of flooding, Figure 7b shows the areas of greatest vulnerability in the center of the city, where in periods of flooding all of the runoff is moved towards that area. Figure 7c shows the vulnerability in those areas that have zero or almost zero soil permeability, such as in streets, commercial and industrial areas, and highly inhabited areas. On the other hand, Figure 7d shows where the medium and high-risk industries have an important influence on the vulnerability due to the productive activities that result in the emission of pollutants into the atmosphere. The integration of all physical layers can be observed in Figure 7e, where the results from the hotspots areas suggest that the areas of greatest vulnerability are those areas where all layers coincide. Finally, the hot spots were mapped for the age-adjusted NCD mortality rates, as shown in Figure 7f. The hotspot analysis identifies one area in the north and one area in northwest of the city with higher risks of death associated with non-communicable diseases.

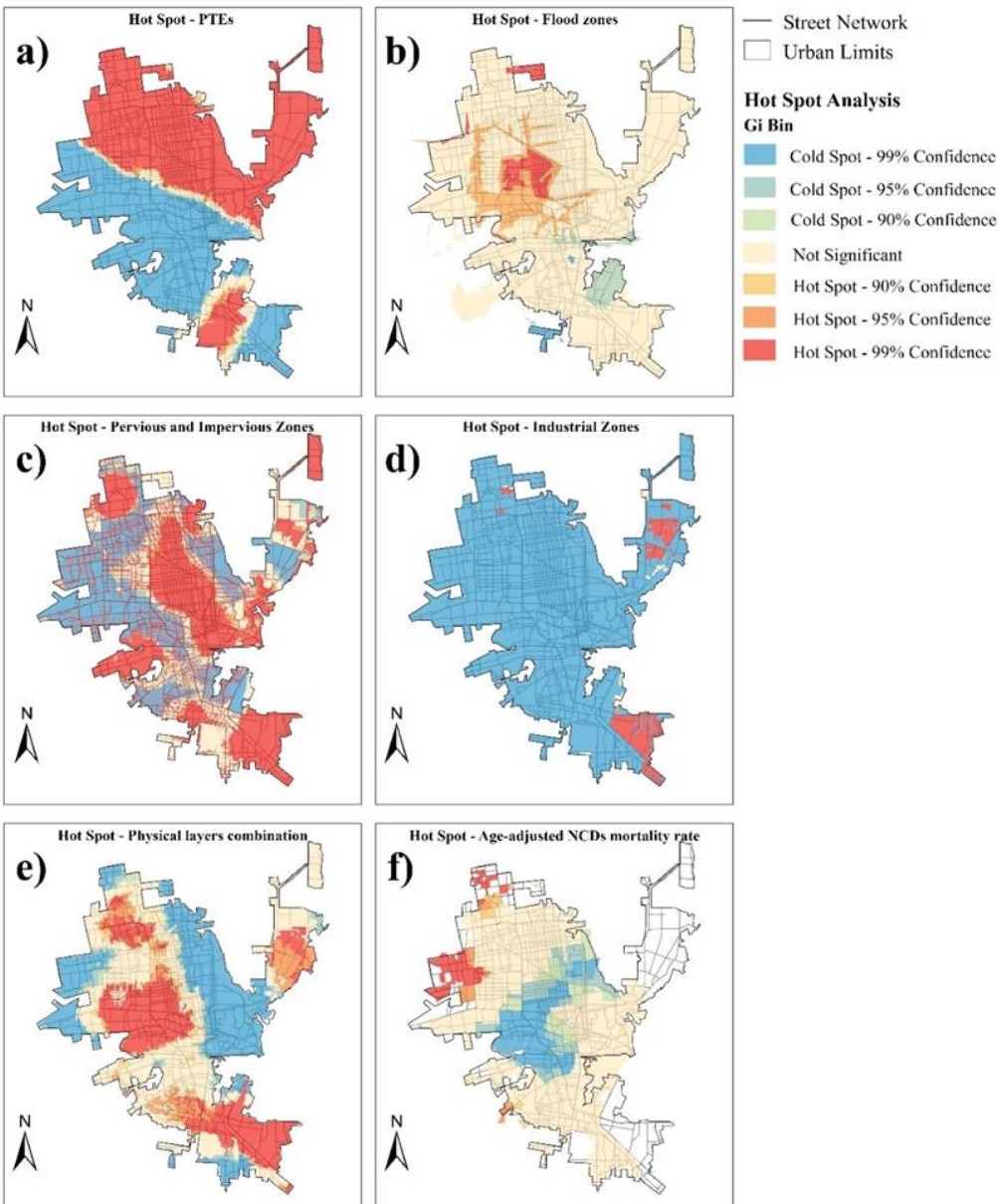

**Figure 7.** Hot spot maps obtained for PTEs and physical layers: (**a**) hot spot for PTEs; (**b**) hot spot for flood zones; (**c**) hot spot for pervious and impervious zones; (**d**) hot spot for industrial zones; (**e**) hot spot for the combination of physical layers; (**f**) hot spot for age-adjusted NCD mortality rates.

Once the base maps were finished, the hot spot maps for the different analyses were obtained. As part of the sensitivity analysis, three maps were chosen as the main maps in this study due to the relationships of the PTEs with physical processes, public health, and their interactions. Figure 8 shows the vulnerability to PTE exposure when integrating physical and public health variables. Figure 8a is the combination of PTEs, flood zones, pervious and impervious areas, and industrial zones. Figure 8b is the result of the combination of the PTEs and the age-adjusted NCD mortality rates. Finally, Figure 8c is one of the most important figures obtained due to its representation of the combination of all variables (PTEs, flood zones, pervious and impervious zones, industrial zones, and the age-adjusted NCD mortality rates).

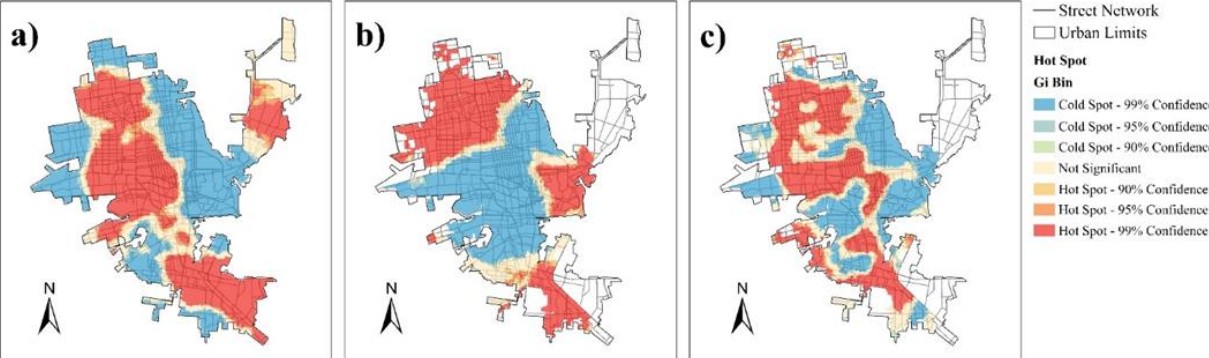

**Figure 8.** (**a**) Vulnerability to PTE exposure when integrating physical variables (flooding zones, impervious zones, and industrial zones). (**b**) Vulnerability to PTE exposure when integrating public health variables (age-adjusted NCD mortality rate). (**c**) Vulnerability to PTE exposure when integrating physical and public health variables (flooding zones, impervious zones, industrial zones, and age-adjusted NCD mortality rates).

### 3.4. Sensitivity Analysis

Once the hot spot maps from the different analyses were obtained, the sensitivity analysis was carried out to determine which variables affect the spatial distribution of cold and hot spots. The product of the sensitivity analysis resulted in a total of 5 variables analyzed individually and 15 different combinations between the flooding, impervious zone, industrial zone, age-adjusted NCD mortality rate, and PTE base maps (Figure 6).

Table 6 represents the hot spot analysis carried out for all base maps without any type of combination, except for the analysis where the flood zones, pervious and impervious zones, and industrial zones were combined. The cold and hot spot areas are shown in Table 6, where the PTEs (48.8%) and the pervious and impervious zones (52.3%) present the highest percentages of vulnerable areas (at 99% confidence).

**Table 6.** Sensitivity analysis of the base maps.

| | | Base Maps | | | | | | |
|---|---|---|---|---|---|---|---|---|
| | | PTEs | Flood Zones | Impervious Zones | Industrial Zones | Combination Physical Layers | Age-Adjusted NCDs Mortality | |
| Gi Bin | Significance | | | | Area | | | Units |
| −3 | Cold Spot-99% Confidence | 41.8 | 0.0 | 24.0 | 93.6 | 29.9 | 12.5 | % |
| −2 | Cold Spot-95% Confidence | 0.7 | 0.8 | 3.3 | 0.004 | 3.3 | 8.2 | % |
| −1 | Cold Spot-90% Confidence | 0.7 | 4.0 | 1.8 | 0.1 | 1.6 | 5.5 | % |
| 0 | Not Significant | 5.8 | 81.3 | 16.3 | 0.5 | 32.3 | 65.3 | % |
| 1 | Hot Spot-90% Confidence | 0.5 | 9.9 | 0.8 | 0.01 | 2.5 | 1.3 | % |
| 2 | Hot Spot-95% Confidence | 1.6 | 3.9 | 1.6 | 0.1 | 8.5 | 2.2 | % |
| 3 | Hot Spot-99% Confidence | 48.8 | 0.0 | 52.3 | 5.6 | 24.9 | 5.2 | % |
| | | 100.0 | 100.0 | 100.0 | 100.0 | 100.0 | 100.0 | % |

In this case, the highest 99% hot spot found was our for PTE map (61.0%), which had previously been referred to as our base map.

On the other hand, Table 7 shows the different combinations between PTEs, physical maps, and public health maps. Analyses 1 to 7 represent the different combinations between the PTEs and the physical maps, where the highest percentages of vulnerable areas (at 99% confidence) were found in analyses 2, 4, and 7, with percentages of 42.7%, 43.1%, and 41.3%, respectively.

**Table 7.** Sensitivity analysis of the different combinations between PTEs, physical maps, and public health maps.

| Gi Bin | Significance | Analysis 1 | Analysis 2 | Analysis 3 | Analysis 4 | Analysis 5 | Analysis 6 | Analysis 7 | Analysis 8 | Analysis 9 | Analysis 10 | Analysis 11 | Analysis 12 | Analysis 13 | Analysis 14 | Analysis 15 | Units |
|---|---|---|---|---|---|---|---|---|---|---|---|---|---|---|---|---|---|
| | | | | | | | | | Area | | | | | | | | |
| −3 | Cold Spot-99% Confidence | 35.8 | 33.2 | 55.5 | 38.3 | 20.1 | 44.4 | 34.5 | 42.1 | 28.6 | 34.9 | 12.0 | 31 | 28.8 | 35.7 | 31.6 | % |
| −2 | Cold Spot-95% Confidence | 2.4 | 2.4 | 1.5 | 2.3 | 3.5 | 2.9 | 2.2 | 1.8 | 5.3 | 2.0 | 1.9 | 3.8 | 5.5 | 1.7 | 3.6 | % |
| −1 | Cold Spot-90% Confidence | 1.3 | 1.2 | 0.7 | 1.2 | 2.5 | 1.5 | 0.9 | 0.8 | 2.7 | 1.0 | 0.9 | 1.9 | 2.6 | 1.0 | 1.9 | % |
| 0 | Not Significant | 17.8 | 15.3 | 8.0 | 11.0 | 21.3 | 21.1 | 15.4 | 11.5 | 26.0 | 15.9 | 12.5 | 20.5 | 27.0 | 16.5 | 20.9 | 5 |
| 1 | Hot Spot-90% Confidence | 2.4 | 2.0 | 0.6 | 1.3 | 2.1 | 2.2 | 1.9 | 1.2 | 3.5 | 2.5 | 1.1 | 1.9 | 2.7 | 2.2 | 2.0 | % |
| 2 | Hot Spot-95% Confidence | 3.7 | 3.2 | 2.0 | 2.7 | 4.0 | 3.0 | 3.8 | 2.6 | 3.9 | 4.2 | 2.1 | 4.4 | 3.9 | 4.2 | 4.2 | % |
| 3 | Hot Spot-99% Confidence | 36.7 | 42.7 | 31.8 | 43.1 | 36.0 | 24.9 | 41.3 | 40 | 30.1 | 39.6 | 39.5 | 36.3 | 29.5 | 38.8 | 35.9 | % |
| | | 100 | 100 | 100 | 100 | 100 | 100 | 100 | 100 | 100 | 100 | 100 | 100 | 100 | 100 | 100 | % |

Analysis 8 represents the combination between the PTEs and the age-adjusted NCD mortality rates, and presents 40% of the total area as vulnerable areas related to mortality and exposure to PTEs. Finally, the different combinations between the PTE, physical, and human health maps are shown from analyses 9 to 15. The analyses that presented greater areas of vulnerability were analyses 10, 11, and 14, with percentages of 39.6%, 39.5%, and 38.8%, respectively.

As a result of the sensitivity analysis, three combinations (analyses 7, 8, and 15) were chosen as representative of the maps that better describe the vulnerability to PTEs in the city. These analyses are characterized by their importance to the distribution and movement of urban dust associated with PTE exposure and its relationship with public health (Table 7). Analysis 7 shows the combination of the spatial distribution of PTEs, floods, pervious and impervious zones, and industrial zones, where the hot spot at 99% confidence represents 41.3% of the total area of Hermosillo. The combination between the PTEs and the age-adjusted NCD mortality rate can be observed in analysis 8, where the hot spot at 99% confidence represents 40.0% of the total area. Finally, analysis 15 shows the combination of PTEs, flood zones, pervious and impervious zones, industrial zones, and the age-adjusted NCD mortality rate, where the hot spot at 99% confidence represents 35.9% of the total area.

## 4. Discussion

### 4.1. Geostatistical Analysis and Spatial Distribution of PTEs

One of the characteristics of geostatistical methods is that the frequency distribution of the data should be close to a normal distribution. However, most applications are not represented using a normal distribution. This is caused by several factors such as the density and sampling scale, which may not be representative, and errors in the laboratory analysis. Therefore, it is necessary to transform the original data to normally distributed data [56]. In soil contamination studies, geostatistical analyses are a powerful tool to separate sources contributing to observed pollution. This technique has been widely used to differentiate between different natural sources that cause variations in soil composition and to identify pollution sources affecting the content of pollutants in the soil [62]. The main application of geostatistics to soil science has been the successful estimation and mapping of soil attributes in unsampled areas [56,62,63]. In our study, the PTE data were processed through a geostatistical analysis to find the best parameters needed to conduct the spatial interpolations. The nuggets and variograms helped us represent the variation in the composition of the elements present in urban dust. In the case of the nugget effect, this refers to the situation in which the difference between measurements taken at sampling locations that are close together is not zero [64]. Other air quality studies have estimated variograms and nuggets, suggesting the use of spherical models with a correlation coefficient greater than 0.90 [18]. On the other hand, the models obtained in our analysis are summarized in Table 5 and are consistent with other studies. For example, Zhang et al. (2014), Li and Feng (2012), and Duan et al. (2015) used multivariate and geostatistical analyses of PTEs found in soil samples, finding that linear, spherical, exponential, and Gaussian models can be fitted adequately to the data [56,62,63]. The models may vary from each other depending on the number and density of samples ($km^2$/sample). A higher density and number of samples can significantly reduce errors and present more stable models [52].

The spatial distribution of PTEs is a powerful indicator that allows by itself for one to identify areas vulnerable to the exposure of these elements. Different sources of pollutant emissions from domestic, commercial, and industrial activities can further concentrate some elements in certain parts of the city, such as Pb and V from combustion exhaust, industrial processes, coal combustion, metallurgic processes, and the construction industry [6,7,65], or chromium (Cr) from traffic paint, which can be liberated by high temperatures and radiation or by the degradation of the asphaltic cover [6].

Meza-Figueroa et al. (2018) reported concentrations of Pb from 21.7 to 778.1 $mg \cdot kg^{-1}$ in road dust that were consistent with our data range of 15.2 to 979.8 $mg \cdot kg^{-1}$ [6]. In the same way, the spatial distribution of Pb presented in their work is consistent with our Pb

distribution map. On the other hand, Cr concentrations can be attributed to metal-based traffic paints on the streets and playground paints [2,6]. Traffic painting releases Cr particles due to the photodegradation process. It has been reported that fresh yellow traffic paint contains high concentrations of Pb and Cr [6]. In this study, higher concentrations of Cr were found in the center of the city (Figure 5b). This can be attributed to the high levels of vehicular traffic and the physical conditions of the asphalt and traffic paint. Meza-Figueroa et al. (2018) also measured the Cr concentrations in road dust in Hermosillo [6]. The Cr concentrations reported in their work ranged from 112.7 to 257.5 mg·kg$^{-1}$, which coincide with our results of 112.6 to 273.5 mg·kg$^{-1}$. The spatial distribution map presented by Meza-Figueroa et al. (2018) is consistent with our distribution map of Cr in Hermosillo.

Vanadium is a toxic metal, which if inhaled can induce pulmonary tumors and increase the probability of lung cancer [61]. The concentrations of V in the ambient air vary widely between rural and urban areas [65]. In urban areas, the V is strategically important due to its wide use in fossil fuels, steel manufacturing, aircrafts, and cement plants [60,61,66]. This could explain the large V distribution in Figure 5c in Hermosillo. Li et al. (2020) measured V in samples of farmland soil in China and the average content reported was 121 mg·kg$^{-1}$ [61]. Our results compared with Li et al. (2020) show higher concentrations with an average of 242.1 mg·kg$^{-1}$. This is mainly due to the difference in climates, whereby Hermosillo, being a semiarid city, tends to accumulate more elements in urban dust.

*4.2. Hot Spots*

Our hot spot results show the spatial distribution of the vulnerability for the different analyses in Hermosillo. In the case of flooding, Figure 7b shows these areas in the center of the city. The vulnerability levels in those areas that have zero or almost zero soil permeability, such as streets, commercial and industrial areas, and highly inhabited areas, are displayed in Figure 7c. On the other hand, Figure 7d shows where the medium- and high-risk industries have an important influence on the vulnerability due to the productive activities that result in the emission of pollutants into the atmosphere. The integration of all physical layers can be observed in Figure 7e, where areas of greater vulnerability are present in those areas that all layers coincide. Finally, a hot spot was mapped for the age-adjusted NCD mortality rate in Figure 7f, where in the north and northwest of the city a higher risk of deaths associated with non-communicable diseases can be observed.

Figure 8 shows the vulnerability to PTE exposure when integrating physical and public health variables. Figure 8a shows the combination of PTEs, flood zones, pervious and impervious zones, and industrial zones. The streets and highly populated areas in Hermosillo have a great influence on the accumulation and transportation of PTEs found in dust. This causes a high vulnerability in those areas that present floods during rain events, influenced by the low or null permeability of the soil.

Figure 8b is the result of the combination of the PTEs and the age-adjusted NCD mortality rate. The vulnerability areas coincide with those areas (west and northwest) where high concentrations of PTEs have been registered, as well as those areas with the highest incidence of mortality that could be associated with the exposure of PTEs.

Figure 8c is one of the most important figures obtained due to its representation of the combination of all variables (PTEs, flood zones, pervious and impervious zones, industrial zones, and the age-adjusted NCD mortality rate). In this case, the areas of greatest vulnerability coincide with those areas where there is a great impact by floods influenced by impervious areas in the city. The transport of urban dust and PTEs during flood events is consistent with the spatial distribution of the PTEs and the areas where deaths that could be related to PTE exposure have been reported.

In the same way as in this work, Navarro-Estupiñan et al. (2020) also used percentage metrics to identify areas vulnerable to heat risk in Hermosillo, Mexico [39]. Zones with high vulnerability were identified in the center of the city, where the hot spots ranged from 1.92 to 36.16% for low- and medium-density housing and mixed areas. Towards the periphery of the city in northern, western, and southern areas, zones with low vulnerability

(cold spots) were identified containing high- and medium-density housing, mixed areas, and housing reserves, ranging from 2.22 to 3.10%. Considering the population percentages of Hermosillo in 2010, based on their results the authors suggested that 16.6% live in areas of high vulnerability, 13.9% in areas of medium vulnerability, and 70.4% in areas of low vulnerability.

### 4.3. Sensitivity Analysis

Once the hot spot maps were obtained, the sensitivity analysis was conducted to identify the layers (variables) with the greatest influence on the spatial distribution of cold and hot spots. Table 6 shows the vulnerable areas considering each variable individually, except for the analysis where the flood zones, pervious and impervious zones, and industrial zones were combined. On the other hand, Table 7 shows the different combinations between the PTEs, physical maps, and public health maps.

The floods and pervious and impervious zones were the variables that most affected the spatial distribution of the hot spots. The hot spot base map (PTEs) at 99% confidence represented 48.8% of the total area. Analysis 3, where the floods and pervious and impervious zones were not used, decreased to 31.8%. Secondly, in analysis 6, where the flood zones were not used, the 99% confidence areas decreased to 24.9%. Therefore, the integration of these variables has a significant influence on the spatial distribution of vulnerable areas.

Analysis 7 shows the combination of the spatial distribution of PTEs and the physical layers, where the hot spot at 99% confidence represents 41.3% of the total area. In analysis 6, where only flood areas are not considered, the area representing hot spots decreases from 41.3% to 24.9%. Therefore, the flood zones were the most sensitive variable due to the weight they had during binary normalization. The combination of the PTEs and the age-adjusted NCD mortality rate can be observed in analysis 8, where the hot spot at 99% confidence represents 40.0% of the total area. Finally, analysis 15 shows the combination of PTEs, physical maps, and the public health layers, where the hot spot at 99% confidence represents 35.9% of the total area. In analyses 9 and 13, where only the permeable and impermeable areas were not considered, the area representing the hot spots decreased from 35.9% to 30.1% and 29.5%, respectively. This means that the permeable and impermeable zones were the most sensitive variables due to the weight they have in the spatial distribution.

Several studies that have tested single and multiple variables using clustering techniques, particularly using the Getis-Ord $G_i$* tool, have reported the use of sensitivity analyses to demonstrate the influence of different variables in their studies. McClintock (2012) performed multiple comparison tests (*z*-scores) in a hot spot analysis to identify soil lead contamination at multiple scales in Oakland, California [35]. This author used land use data, soil Pb concentrations from 112 sites (ranged from 3 to 979 mg·kg$^{-1}$), types of vegetation, and variations in geographical zones (neighborhood-scale) to evaluate the risk of Pb contamination. In this study, a range of *z*-scores from $-2.58$ to $2.58$ was found, where a high *z*-score indicates the clustering of high soil Pb concentrations, while a low *z*-score indicates the spatial clustering of low Pb concentrations. Median *z*-scores suggest that there is no significant spatial relationship between the Pb concentrations and the geographical zones. A Getis Ord $G_i$* test on the neighborhoods-scale data revealed the significant clustering (hot spot) of elevated Pb concentrations in the southwest corner of West Oakland. Similarly, Lee and Khattak (2019) used *z*-scores and *p*-values to determine differences in spatial clusters or hot spot areas of crash points on roadway networks in Lincoln, Nebraska [34]. Eight high-severity crash clusters and six low-severity clusters were identified in the study area. The clusters were determined based on the number of statistically significant crash points (at least eight points in a cluster) and their significance level ($p \leq 0.01$).

Cooper-Vince et al. (2018) used *p*-values (Poisson regression) to identify the risk of depression associated with water insecurity, gender, marital status, education, assets of wealth, and overall health using a hot spot analysis in a rural parish in Mbarara District, Uganda [37]. The results of the sensitive analysis suggest that women who reside in a water

insecurity hot spot have a 70% higher risk ($p = 0.003$) of probable depression compared with women who do not reside in a water insecurity hot spot. On the other hand, men who reside in a water insecurity hot spot do not have a risk of probable depression ($p = 0.92$); however, with the multiple regression model, the interaction between gender (men and women) and living in a water insecurity hot spot was not statistically significant ($p = 0.08$). The results of the sensitivity analysis suggest that education level, age, marital status, wealth, and general health are not significant factors in depression. In this case, gender (female) and areas of water insecurity were the most important factors in determining depression.

The sensitivity analysis carried out by García et al. (2018) to evaluate water infrastructure failures in three cities in California used multivariate linear regression models ($p$-values) to determine significant differences between different variables (the pipe material, season, diameter, and soil types) and across the cities [67]. The sensitivity analysis suggested that the selected pipe material, season, diameter, and soil types have statistically significant ($p < 0.05$) effects on the pipe longevity.

Another study conducted by Zhang and Tripathi (2018) used a Pearson correlation ($p$-value) in a hot spot analysis to investigate lung cancer and its spatial correlation to mortality and fine particulate matter ($PM_{2.5}$) using data from 2008 to 2012 [38]. In this case, the age-standardized incidence rate (ASR) and age-standardized mortality rate (ASMR) of lung cancer were closely correlated with the $PM_{2.5}$ value ($p \leq 0.01$). A second correlation was performed to determine the correlations between lung cancer, $PM_{2.5}$, wind speed, and wind direction. The results suggested that the wind direction is an important factor that affects the $PM_{2.5}$ value ($p \leq 0.01$).

Finally, Navarro-Estupiñan et al. (2020) performed a sensitive metric analysis (%) using thermal maps, socioeconomic data (i.e., gender, age, marital status, education level, health services), and physical indicators (housing with electricity, fridge and washing machine, Internet access, and phone and cellphone, as well as impervious areas and streets) to determine the differences in hot spot areas in different analyses compared with the hot spot map with the analysis of all indicators in a semiarid city in Northwestern Mexico [39]. The most sensitive indicators were found to be age and education ($\geq$15 years old without elementary school), with a 2.0% difference in hot spot areas, followed by health (without health public service) at a 1.6% difference and age (18–65 years) at a 1.1% difference. These indicators are related to heat exposure through outdoor activities such as construction works and agricultural activities in which people participate daily for economic reasons. Although the results presented by Navarro-Estupiñan et al. (2020) do not evaluate the same variables as in our study, their results can be compared in terms of the units [39]. The work proposed by Navarro-Estupiñan et al. (2020) determined the differences in the percentages of areas presented in the different analyses, such as for age and education and health and age [39]. Our results in the same way determined the most sensitive variables in the distribution of vulnerable areas, as in the case of flood zones and permeable and impermeable zones (analysis 3) with a difference of 17% in more vulnerable areas. In the specific case of flooded zones (analysis 6), this variable showed a difference of 23.9%, being the most sensitive compared with the base map (PTEs).

The results presented in this work are not directly comparable to the other studies because all have different contexts, with the exception of the work conducted by Navarro-Estupiñan et al. (2020) [39]. In our study, we were able to highlight the most dynamic variables after they were grouped with variables that do not change much over time. This procedure helped identify a more robust measure of vulnerability to PTEs. We argue that it is critical to conduct a sensitivity analysis to quantify the influences of different layers. Previous studies used statistical tools to highlight the influence of the variables used in their studies, which are valid tools that demonstrate the sensitivity of their variables. However, in our case, we quantified the spatial distribution changes in the final maps of vulnerability. We believe that this quantifiable spatial analysis of the increase or decrease in cold and hot spots provides a clearer picture of the most sensitive variables in terms of the influence on the overall vulnerability.

## 5. Conclusions

This work shows that by integrating multiple layers with field observations of PTEs found in urban dust, a more robust vulnerability map was developed, which can be helpful for urban planning and civil protection agencies. The sensitivity analysis proved to be an important tool to identify the variables that have different levels of influence on the extent of the distribution of vulnerable areas. In general, in this study we found that the flood zones and permeable and impermeable areas of a semiarid city have the highest importance in relation to the overall vulnerable areas. The absence of these variables in the different analyses caused the loss of spatial variability.

We argue that in the analysis of PTEs and cities that have a high influence from seasonal monsoon systems, it is difficult to use only dust samples to determine areas of vulnerability because of the dynamic conditions that affect the distribution of the dust. For this reason, it is important to integrate covariates (wind speed and direction, topography, etc.) that help model the spatial distribution of the elements in a more robust way. In the same way, the integration of physical and public health variables to obtain vulnerability maps is of valuable importance to understand the areas that are more vulnerable.

The clustering techniques such as the hot spots used here are proven techniques that can help point out areas that are critical for the design of strategies aimed at reducing exposure to environmentally related public health risks. However, the hot spots maps can be improved significantly by adding multiple layers and quantifying their impact on the overall vulnerability map.

In this paper, we sought to provide a road map for other cities in using available spatially explicit information to highlight public and environmental health risks. We also aimed to provide important information regarding environmental justice problems. We hope that the information presented here can also be useful for non-government organizations that are providing information to the public.

**Supplementary Materials:** The following supporting information can be downloaded at: https://www.mdpi.com/article/10.3390/su141710461/s1, Figures S1–S14: Spatial distribution of Ba, Ca, Cr, Cu, Fe, Pb, Nb, K, Sr, Th, Ti, V, Y and Zn in Hermosillo; Figures S15–S29: show the individual sensitivity analyzes of hot spots; Tables S1–S9: show the percentage of area for cold and hot spots.

**Author Contributions:** Conceptualization, E.V.-J., A.R.-M. and D.M.-F.; methodology, E.V.-J.; software, E.V.-J.; validation, E.V.-J.; formal analysis, E.V.-J.; investigation, E.V.-J., A.R.-M., D.M-F. and P.A.R.-C.; resources, A.R.-M. and D.M.-F.; data curation, E.V.-J.; writing—original draft preparation, E.V.-J.; writing—review and editing, A.R.-M.; visualization, E.V.-J.; supervision, A.R.-M. and D.M.-F.; project administration, A.R.-M.; funding acquisition, A.R.-M. and D.M.-F. All authors have read and agreed to the published version of the manuscript.

**Funding:** This research was funded by Instituto Tecnológico de Sonora (TSON), Programa de Fomento y Apoyo a Proyectos de Investigacion (PROFAPI), and Laboratorio Nacional de Geoquímica y Mineralogía (LANGEM). This material was also supported by the Urban Resilience to Extreme Events Sustainability Research Network of the National Science Foundation under award number: SES-1444755.

**Institutional Review Board Statement:** This study did not require ethical approval.

**Informed Consent Statement:** Not applicable.

**Data Availability Statement:** The PTE data used in this study was provided by Dr. Meza-Figueroa (co-author). The Age-adjusted NCD mortality rates map was provided by Dr. Reyes-Castro (co-author). All the other GIS data was generated by the first author using publicly available data described in the methodology section.

**Acknowledgments:** We acknowledge the support of CONACYT for Efrain Vizuete-Jaramillo through a graduate fellowship. The support of the Instituto Municipal de Ecologia (IMEC) and Universidad de Sonora (UNISON) is also appreciated.

**Conflicts of Interest:** The authors declare no conflict of interest.

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
