# Peer review of "Using a Sensitivity Analysis and Spatial Clustering to Determine Vulnerability to Potentially Toxic Elements in a Semiarid City in Northwest Mexico"

_sustainability, doi:10.3390/su141710461_

Round 1

Reviewer 1 Report

The results presented by the authors are interesting. Proper use of statistical tools is made. I think the relationship between air quality and the presence of PTEs and health problems can be better discussed. Some recommendations to improve the manuscript are indicated in the text.

Author Response

Thank you for taking the time to review our manuscript. Attached is our file with the responses to each of your comments.

Reviewer 2 Report

General comments:

This is an interesting manuscript. The authors combined multiple variables and implemented a clustering technique to create a hot spot exposure map for 14 Potentially Toxic Elements (PTEs) found in urban dusts in the city of Hermosillo, Mexico. In my opinion some issues need to be clarified. Please see the specific comments section of my review.

Specific comments:

Lines 57-59: This section is repetitive and confusing. Please delete it.

Lines 62-64: The expression “grouping similar objects into different groups” is not clear. Please revise and keep only the second definition for cluster analysis.

Lines 86; 92; etc: The references are included without the year of publication. Is this correct?

Lines 189-190: Why the dust samples were not analysed for other important PTEs such as As, Cd and Ni? This must be explained.

Line 329: The content of the parenthesis is confusing at this spot. Please revise.

Lines 375-376: Use subscripts in order to match the text content with the symbols in equation 2.

Lines 426-427: Vanadium is a marker of heavy fuel oil combustion especially from shipping and not a marker of air pollution in general. Please revise and include more information and more references.

Lines 432-433: Concentrations exceeding the background values are not necessarily associated with enhanced health risks. Please revise or delete this phrase.

Lines 445-448: The legend is confusing. Please revise.

Lines 463-466: The authors say that at the east of the city there is a higher risk of deaths associated with non-communicable diseases. I don’t see that in Figure 7f. In addition, the hotspot located at the Northwest part of the city is not highlighted in the previous maps and especially in Figure 7e where all physical layers are combined. The authors must revise this section and provide an explanation.

Lines 478-479: “pervious and impervious” is repeated. Please revise.

Figure 8c: Again, in this important figure the primary mortality hotspot located at the Northwest part of the city is not highlighted. Please explain.

Lines 498-522 and Table 7. More information regarding all and not only some of the different analyses are needed here or at least in a table included at the supplementary material. Only the maps are not enough.

Lines 591-592: “pervious and impervious” is repeated again. Please revise.

Lines 599-600: The statement that “There is no direct link that NCDs mortality in Hermosillo was caused by PTEs” is very important and must be further explained. As I previously mentioned the NCDs mortality hotspot located at the Northwest part of the city is not in good agreement with the hot spot maps created for PTEs and physical layers.

Lines 608-610: As I previously mentioned in Figure 8c the primary mortality hotspot located at the Northwest part of the city is not highlighted. Please explain.

Lines 645-718: Most references in this section are irrelevant. Please delete them and revise the whole section.

Author Response

Thank for your comments and for taking the time to review our manuscript. 

Round 2

Reviewer 2 Report

The authors of the manuscript entitled “Using sensitivity analysis and spatial clustering to determine vulnerability to potentially toxic elements in a semiarid city in northwest Mexico” have performed the requested changes in their manuscript, therefore I believe that now it can be accepted for publication in Sustainability journal.